# SARS-CoV-2 E and 3a Proteins Are Inducers of Pannexin Currents

**DOI:** 10.3390/cells12111474

**Published:** 2023-05-25

**Authors:** Barbara B. R. Oliveira-Mendes, Malak Alameh, Béatrice Ollivier, Jérôme Montnach, Nicolas Bidère, Frédérique Souazé, Nicolas Escriou, Flavien Charpentier, Isabelle Baró, Michel De Waard, Gildas Loussouarn

**Affiliations:** 1L’institut du Thorax, Nantes Université, CNRS, INSERM, F-44000 Nantes, France; barbara.ribeiro@univ-nantes.fr (B.B.R.O.-M.); malak.alameh@univ-nantes.fr (M.A.);; 2Labex Ion Channels, Science and Therapeutics, F-06560 Valbonne, France; 3Team SOAP, CRCI2NA, INSERM, CNRS, Nantes Université, Université d’Angers, F-44000 Nantes, France; 4Equipe Labellisée Ligue Contre le Cancer, F-75006 Paris, France; 5CRCI2NA INSERM, CNRS, Nantes Université, F-44000 Nantes, France; 6Institut Pasteur, Université Paris Cité, Département de Santé Globale, F-75015 Paris, France

**Keywords:** COVID-19, SARS-CoV-2, viroporins, E protein, 3a protein, pannexin currents, cell death

## Abstract

Controversial reports have suggested that SARS-CoV E and 3a proteins are plasma membrane viroporins. Here, we aimed at better characterizing the cellular responses induced by these proteins. First, we show that expression of SARS-CoV-2 E or 3a protein in CHO cells gives rise to cells with newly acquired round shapes that detach from the Petri dish. This suggests that cell death is induced upon expression of E or 3a protein. We confirmed this by using flow cytometry. In adhering cells expressing E or 3a protein, the whole-cell currents were not different from those of the control, suggesting that E and 3a proteins are not plasma membrane viroporins. In contrast, recording the currents on detached cells uncovered outwardly rectifying currents much larger than those observed in the control. We illustrate for the first time that carbenoxolone and probenecid block these outwardly rectifying currents; thus, these currents are most probably conducted by pannexin channels that are activated by cell morphology changes and also potentially by cell death. The truncation of C-terminal PDZ binding motifs reduces the proportion of dying cells but does not prevent these outwardly rectifying currents. This suggests distinct pathways for the induction of these cellular events by the two proteins. We conclude that SARS-CoV-2 E and 3a proteins are not viroporins expressed at the plasma membrane.

## 1. Introduction

SARS-CoV-2 is the third virus of the genus Beta-coronavirus of the Coronaviridae family to be responsible for a Severe Acute Respiratory Syndrome in this century, after SARS-CoV-1 in 2002–2003 [1] and MERS-CoV in 2012 [2]. As a result, it is of great importance to best characterize the coronaviruses and their associated pathophysiologies, with the hope that new treatments will emerge to complement vaccine approaches for people who cannot access the vaccines or are not responsive to them. In addition to Paxlovid which is already available but associated with bothersome side-effects [3], many potential anti-COVID-19 treatments are in development, but it is too soon to tell how efficient they will be, namely with regard to the continuous emergence of new variants, and if their cost will be reasonable [4].

Viroporins, i.e., ion channels encoded by a virus genome, are potential targets for antiviral agents, as demonstrated by the case of amantadine, which inhibits the acid-activated M2 channel of the Influenza A virus [5]. Several studies led to the suggestion that two proteins of SARS-CoV are viroporins. The SARS-CoV-2 Envelop (E) protein is a one-transmembrane-domain membrane protein (75 amino-acids) almost identical to the SARS-CoV-1 Envelop protein (95% identity). The SARS-CoV-2 ORF3a (3a) protein is a larger three-transmembrane-domain membrane protein (275 amino-acids) relatively similar to the SARS-CoV-1 3a protein (73% identity).

Regarding the ion channel function of these proteins, there are clearly several contradicting studies: some of them raising intriguing issues, while others do not confirm these reports. Concerning in vitro membrane incorporation of purified E or 3a protein in lipid bilayers, the presence of ion channel activity is reportedly associated with these viral proteins [6,7,8,9,10,11]. However, a review article soundly outlined the lack of robust data and raised ethical concerns, casting doubts on the validity of these scientific messages [12]. Concerning viral protein expression in cells, the expression of SARS-CoV-1 E protein also led to conflicting results [13,14]. Pervushin et al. managed to identify plasma membrane currents generated by heterologous expression of SARS-CoV-1 E protein in HEK-293 cells [13], but not Nieto-Torres et al. [14]. In Pervushin’s study, the strongest evidence for E protein expression at the plasma membrane forming ion channels was the finding that (i) hexamethylene amiloride (HMA) inhibits the induced current and (ii) directly binds to the E protein. A recent study also detected current after injection in *Xenopus laevis* oocytes of any RNA among four different RNAs encoding SARS-CoV-2 proteins, including the E protein [15]. On the other hand, in other studies, expression of SARS-CoV-2 E protein did not lead to interpretable ionic currents in HEK-293S cells or *Xenopus laevis* oocytes [16,17]. In an attempt to favor plasma membrane targeting and reveal a putative current, a C-terminal predicted ER retention signal of SARS-CoV-2 E protein was replaced by a Golgi export signal from the Kir2.1 channel. The expression of this chimera could then be associated with the generation of a non-rectifying and cation-selective current [16]. This current was thus quite different from the outwardly rectifying current observed by Pervushin and collaborators [13]. Furthermore, another study using a membrane targeting sequence, fused to the N-terminus of the SARS-CoV-1 E protein, provided a non-rectifying current that was 100-fold larger than the one observed in the two previous studies [18]. This suggests that such modifications of either the N- or C-terminus are too drastic to faithfully report the actual activity of the native proteins.

The SARS-CoV 3a protein was also investigated. Confocal immuno-imaging detected the expression of WT 3a protein in both plasma membrane and cytoplasm. Membrane expression was reduced for a mutant that showed less current when expressed in HEK-293 cells [19]. Expression of the wild-type (WT) protein in HEK-293 cells but also *Xenopus laevis* oocytes was associated with a poorly selective outwardly rectifying current in both models, resembling the one observed upon expression of the E protein [15,20,21,22]. However, again, these observations were not replicated by other laboratories [23].

To summarize, there is no unequivocal evidence that SARS-CoV E and 3a proteins are viroporins active at the plasma membrane of the host cell. However, on one hand, it was recently reported that SARS-CoV-2 E and 3a proteins can promote cell death [24,25]. On the other hand, apoptosis is associated with an increase in the outwardly rectifying current conducted by pannexins [26,27,28] and VRAC channels [29]. This led us to reinvestigate the actual function(s) of SARS-CoV-2 E and 3a proteins in mammalian cells in the frame of the cell toxicity of these proteins.

In this study, CHO cells expressing either CoV-2 E or 3a proteins tended to develop a round-shaped form with a tendency to detach from the Petri dish, a process exacerbated compared to control conditions. This cell phenotype is consistent with cell death [30,31] and we confirmed via flow cytometry experiments that expression of E or 3a proteins does indeed promote cell death. Transfected cells, still attached to the Petri dish (adhering cells), had unchanged basal currents, indicating that E and 3a proteins are unlikely to act as plasma membrane channels. In contrast, recording whole-cell currents on round-shaped and detached cells, we observed large outwardly rectifying currents only in E or 3a protein-expressing cells but not in control dying cells. This current is reminiscent of those observed in previous publications using HEK-293 cells and oocytes expressing SARS-CoV-1 proteins [13,20,21,22]. The application of carbenoxolone and probenecid, two inhibitors of pannexin channels, suggests for the first time that these currents are pannexin-mediated conductances, potentially activated by altered morphology or apoptosis. In conclusion, both SARS-CoV-2 E and 3a proteins are most likely triggers of endogenous conductance.

## 2. Materials and Methods

### 2.1. Cell Culture

The Chinese Hamster Ovary cell line, CHO, was obtained from the American Type Culture Collection (CRL-11965, Manassas, VA, USA) and cultured in Dulbecco’s modified Eagle’s medium (Gibco 41966-029, Gaithersburg, MD, USA) supplemented with 10% fetal calf serum (Eurobio, Les Ulis, France), 2 mM L-Glutamine and antibiotics (100 U/mL penicillin and 100 µg/mL streptomycin, Corning, NY, USA) in 5% CO_2_, maintained at 37 °C in a humidified incubator. This cell line was confirmed to be mycoplasma-free (MycoAlert, Lonza, Basel, Switzerland).

### 2.2. Drugs

Carbenoxolone disodium salt was purchased from Sigma (St. Louis, MO, USA), and a 100 mmol/L stock solution was prepared in H_2_O. Probenecid was purchased from Sigma, and a 200 mmol/L stock solution was prepared in DMSO. The drugs used for the flow cytometry experiments were QVD-OPh (#OPH001, R&D Systems, Minneapolis, MN, USA, 10 mmol/L stock solution in DMSO), S63-845 (Chemietek, Indianapolis, IN, USA, 10 mmol/L stock solution in DMSO), and ABT737 (Selleckchem, Houston, TX, USA, 10 mmol/L stock solution in DMSO).

### 2.3. Construction of E and 3a Protein-Encoding Plasmids

SARS-CoV-2 E and 3a nucleotide sequences containing a Kozak sequence added right before the ATG (RefSeq NC_045512.2) were synthesized by Eurofins (Ebersberg, Germany) and subcloned into the pIRES2 vector with eGFP in the second cassette (Takara Bio Europe, Saint-Germain-en-Laye, France). Truncated ∆4 and ∆8 E proteins as well as ∆10 3a protein constructs lacking the last 12, 24, and 30 nucleotides, respectively, were also synthesized by Eurofins. The plasmid cDNAs were systematically re-sequenced by Eurofins after each plasmid in-house midiprep (Qiagen, Hilden, Germany).

### 2.4. E and 3a cDNA Transfection

Fugene 6 transfection reagent (Promega, Madison, WI, USA) was used to transfect WT and mutant E and 3a plasmids for patch clamp, morphology analysis, and flow cytometry experiments according to the manufacturer’s protocol. The cells were cultured in 35 mm dishes and transfected at 20% confluence for patch clamp experiments and 50% confluence for flow cytometry assays, with a pIRES plasmid (2 µg DNA) with the first cassette empty or containing wild-type or truncated SARS-CoV-2 E or 3a protein sequences. For morphology analysis, cells were cultured in ibidi μ-Slide 8-well dishes and transfected at 20% confluence with the same plasmids. In pIRES2-eGFP plasmids, the second cassette (eGFP) is less expressed than the first cassette, guaranteeing the expression of a high level of the protein of interest in fluorescent cells [32,33].

### 2.5. Electrophysiology

Two days after transfection, the CHO cells were mounted on the stage of an inverted microscope and bathed with a Tyrode solution (in mmol/L: NaCl 145, KCl 4, MgCl_2_ 1, CaCl_2_ 1, HEPES 5, glucose 5, pH adjusted to 7.4 with NaOH) maintained at 22.0 ± 2.0 °C. Patch pipettes (tip resistance: 2.0 to 2.5 MΩ) were pulled from soda-lime glass capillaries (Kimble-Chase, Vineland, NJ, USA) with a Sutter P-30 puller (Novato, CA, USA). A fluorescent cell was selected via epifluorescence. The pipette was filled with intracellular medium containing (in mmol/L): KCl, 100; Kgluconate, 45; MgCl_2_, 1; EGTA, 5; HEPES, 10; pH adjusted to 7.2 with KOH. Stimulation and data recording were performed with pClamp 10, an A/D converter (Digidata 1440A), and a Multiclamp 700B (all Molecular Devices, San Jose, CA, USA) or a VE-2 patch-clamp amplifier (Alembic Instruments, Montreal, QC, Canada). The currents were acquired in the whole-cell configuration, low-pass filtered at 10 kHz and recorded at a sampling rate of 50 kHz. First, a series of twenty 30-ms steps to −80 mV was applied using alternating holding potential (HP) values of −70 mV and −90 mV, and Cm and Rs values were subsequently calculated offline from the recorded currents. The currents were then recorded using a 1-s ramp protocol from −80 mV to +70 mV every 4 s. Regarding non-adhering cells, we considered them as having large current density when the current density measured at +70 mV was superior to mean + 2 × standard deviation of the current density in adhering cells in the same condition.

### 2.6. Cell Morphology Assay

Cell roundness was estimated using the *Analyze Particle* function of the Fiji software (v 1.53), as described in Appendix A.

### 2.7. Flow Cytometry Assay

Two days after transfection, the CHO cells were prepared for cell death detection following the user guide (https://assets.thermofisher.com/TFS-Assets/LSG/manuals/mp13199.pdf, accessed on 2 February 2021) to measure annexin V binding and propidium iodide (PI) intake. The cells were washed with cold PBS, trypsinized, collected via centrifugation and gently resuspended in annexin-binding buffer (V13246, Invitrogen, Carlsbad, CA, USA) at 1 × 10^6^ cells/mL. To each 300 µL cell suspension were added 0.5 µL of annexin V AlexaFluor 647 (A23204, Invitrogen, Carlsbad, CA, USA) and 1 µL of propidium iodide (PI) at 100 µg/mL (P3566, Invitrogen, Carlsbad, CA, USA). The CHO cells were incubated at room temperature for 15 min in the dark, then maintained on ice until flow cytometry analysis within one hour.

To study the cell death pathways induced by E and 3a protein expression, non-transfected or transfected CHO cells were treated with inhibitors or inducers of apoptosis (inhibitor: 5 µmol/L QVD-OPh incubated for 48 h; activators: 3 µmol/L S63-845 + 8 µmol/L ABT737 incubated for 3 h).

The cytometer BD FACSCanto (BD Biosciences, Franklin Lakes, NJ, USA) was used to sample acquisition. CHO cells transfected with an empty plasmid were used to determine the population to be analyzed. Monolabeled cells were used to establish the photomultiplier voltage of each channel (PMT) and proceed with fluorescence compensation after the acquisitions. In order to detect cell death, only eGFP-positive CHO cells (FITC) were selected to study Annexin V AlexaFluor 647 (APC) and PI (Perc-P) labeling. The analyses were performed using FlowJo software (v10.7.1).

## 3. Results

We first focused on native E and 3a proteins. To maximize the chance of observing E and 3a protein-induced ionic currents, we chose to use pIRES plasmids, in which the protein of interest situated in the first cassette is more expressed than the eGFP reporter in the second cassette, thereby guaranteeing the expression of a high level of the protein of interest in fluorescent cells [32,33]. For the purpose of this study, we also selected CHO rather than HEK-293 cells because they express minimal endogenous currents [34]. We compared whole-cell currents recorded during a ramp protocol in cells transfected either with a control pIRES2-eGFP plasmid (pIRES) or the same plasmid containing the cDNA of the SARS-CoV-2 E protein (pIRES–E) or 3a protein (pIRES–3a). Unexpectedly, we did not observe any difference in the currents recorded for the SARS-CoV-2 protein-expressing cells compared to the control pIRES condition (Figure 1A). However, many cells transfected with either E- or 3a-encoding plasmids developed altered morphology, shifting from spindle-like cells to more round cells (Figure 1B), similar to what was previously observed in MDCK cells heterologously expressing SARS-CoV-1 E protein [35]. Morphology analysis with a Fiji tool confirmed an increase in cell roundness (Figure 1C and Appendix A). In particular, in the patch clamp experiments, some cells were coming off from the dish bottom because of loss of adhesion. Cell counting indicated that slightly more cells were losing adhesion when E or 3a proteins were expressed (3.4 ± 0.6% in non-transfected cells, 5.2 ± 1.0% in pIRES condition, 6.6 ± 0.7% in pIRES–E, *p* < 0.001 vs. pIRES, 6.0 ± 1.2% in pIRES–3a, *p* < 0.001 vs. pIRES, five replicates, *z*-test). As is standard, the currents shown in Figure 1A were recorded from the adhering cells, while the non-adhering cells were disregarded in this initial investigation. Noteworthily, in each condition, both spindle-like and round adhering cells were studied (pIRES: 21 spindle-like and 17 round cells; pIRES E: 9 and 13, pIRES 3a: 18 and 11).

Since both E and 3a proteins promote cell death [24,25], we hypothesized that the various cell morphological patterns (spindle-shaped, round-adhering, and round non-adhering) may correspond to the development of cell death, as described earlier in CHO and other cells [30,31]. The flow cytometry analysis performed on the eGFP-positive CHO cells (Figure 2) showed that expression of E and 3a proteins increases the percentage of dying cells, with more significantly late cell death, revealed by propidium iodide permeability (Appendix A). The effect of 3a protein was greater than the effect of E protein. E protein-induced cell death could be reduced by the pan-caspase inhibitor QVD-OPh, while 3a protein-induced cell death could not (Appendix A), suggesting that E protein induces apoptosis, while 3a protein activates non-conventional caspase-independent cell death.

Both E and 3a proteins possess a C-terminal PDZ binding motif (PBM). E-protein PBM has been suggested to be a virulence factor [11] that binds to host cell PDZ domains, leading to abnormal cellular distribution of the bound proteins [35]. 3a PBM interacts with at least five human PDZ-containing proteins (TJP1, NHERF3 and 4, RGS3, PARD3B), suggesting that it also alters cellular organization [36]. We thus evaluated whether deletion of these domains impacts the propensity of E and 3a proteins to trigger cell death. Two C-terminal deletions used in previous studies to remove E protein PBM, ∆4 for the last four amino acids [35] and ∆8 for the last eight residues [11], abolished the pro-apoptotic effect of E protein (Figure 2). When looking individually at early and late cell death, we observed that both truncations of E and 3a protein decreased late cell death (Appendix A).

Since both E and 3a proteins promote altered morphology and cell death, we hypothesized that the cells starting to come off the surface may express currents induced by altered morphology and/or cell death, such as volume-regulated anion channel (VRAC) or pannexin currents [26,27,28,29,37,38]. We thus compared patch clamp recordings of adhering cells vs. non-adhering cells for three conditions: control pIRES, pIRES–E and pIRES–3a plasmids (Figure 3). For the control pIRES condition, focusing on non-adhering cells in the 35 mm dish and using the ramp protocol, we observed an outwardly rectifying current with a mean current density of 8.5 ± 3.1 pA/pF at +70 mV, slightly higher than those of spindle- or round-shaped adhering cells (3.4 ± 0.8 pA/pF). On the other hand, in non-adhering cells expressing either the E or 3a protein, currents were much larger in the E protein condition (I_+70mV_ = 31 ± 9 pA/pF, two-way ANOVA test on the ramp-evoked currents: *p* < 0.0001) and the 3a protein condition (I_+70mV_ = 44 ± 13 pA/pF, two-way ANOVA test on the ramp-evoked currents: *p* < 0.0001) compared to non-adhering cells in the control pIRES condition. Noteworthily, only a fraction of the non-adhering cells exhibited large rectifying currents, as shown in Figure 3: 4 out of 43 in the control pIRES condition, 14 out of 46 in the E protein condition, and 16 out of 41 in the 3a protein condition. These experiments suggest that changes in morphology and/or cell death induced by expression of E and 3a proteins may lead to an increased membrane permeability by enhancing the expression or activity of an endogenous ion channel.

Current density is classically used to compare channel activity in single cells, but since cell morphology is affected by the expression of E and 3a proteins, we tested if membrane capacitance is modified in non-adhering cells. Appendix A actually shows a reduction in membrane capacitance in non-adhering cells in several conditions. In order to verify that the increase in current density observed in non-adhering cells in Figure 3 is not indirectly due to the decrease in membrane capacitance, we also compared the current amplitudes (not divided by membrane capacitance) and still observed significantly larger rectifying currents when E or 3a protein was expressed (Appendix A).

The outwardly rectifying currents that we observed resemble both VRAC currents conducted by swelling activated anion channels and pannexin currents that are not only apoptosis-induced but also stretch-induced [26,27,28,29,37,38]. We chose carbenoxolone (CBX), which inhibits both channels with similar affinity [39,40], and applied it on non-adhering cells that display large outwardly rectifying currents (Figure 4). We observed that CBX, applied at 50 µmol/L, inhibits the observed current, restoring current amplitudes similar to the ones observed in the control cells. Probenecid is commonly used to inhibit pannexin channels and seems quite specific for pannexin currents, showing little effect on connexin channels and no described effect on VRAC channels [40,41]. We observed that probenecid, applied at 300 µmol/L, also inhibits the large outwardly rectifying currents observed in the non-adhering E protein expressing cells (Figure 5). Altogether, these observations suggest that the current triggered by the expression of E and 3a proteins is most probably conducted by pannexin channels.

We reported above (Figure 2) that deleting the last four amino acids of the E protein (∆4) drastically reduced its pro-apoptotic effect. Cells expressing the ∆4 E protein showed an average roundness similar to cells expressing the WT protein, suggesting that deletion did not prevent its effect on cell morphology (Figure 6A and Appendix A). Additionally, when focusing on round and non-adhering ∆4 E protein-expressing cells, we could still record large outwardly rectifying currents (5 out of 20 cells), suggesting that C-terminal deletion of E protein does not abolish the induction of pannexin currents, despite the prevention of apoptosis probed by the flow cytometry experiments (Figure 6B and Appendix A).

We also reported in Figure 2 that the deletion of the last 10 amino acids of the 3a protein (∆10) also decreased cell death, albeit to a lesser extent. As for E protein deletion, cells expressing the truncated 3a protein showed an average roundness similar to cells expressing the WT protein (Figure 6A and Appendix A). Focusing on the non-adhering cells, we could still record large outwardly rectifying currents (9 out of 14 cells), suggesting that deletion of the last 10 amino acids is not sufficient to abolish the induction of pannexin-like currents (Figure 6C and Appendix A).

## 4. Discussion

The concept that SARS-CoV E or 3a proteins could be viroporins expressed at the plasma membrane is a seductive one, as it could help the identification of new therapeutic drugs against COVID-19 by setting up a screening program based on channel activity. However, this concept is controversial, and some of the reasons that explain this controversy about the function of E and 3a proteins is likely linked to the fact that these proteins also trigger morphological alterations and/or cell death. One could imagine for instance that morphological alterations and/or cell death would be a way to activate the function of E and 3a proteins at the plasma membrane, but an alternative hypothesis could be simply that morphological alterations and/or cell death triggers endogenous cell conductances unrelated to the cell function of E and 3a viral proteins [26,27,28,29,38,42].

The only way to address these issues is to confirm that both morphological alterations and cell death are induced by E and 3a proteins, to measure plasma membrane conductance, and to characterize them to get insights on their nature and probe the pharmacological agents that would match their conductance identities. We managed to solve these issues by characterizing the membrane conductances triggered by both E and 3a proteins. The fact that both proteins trigger the same conductance independently of each other was the first indication that they could not be viroporins at the plasma membrane. The second hint was that adhering cells, whether they had a round shape or not, did not exhibit any outward conductance in spite of E or 3a protein expression. Finally, the sensitivity to carbenoxolone of the outwardly rectifying currents triggered by E or 3a proteins in non-adhering cells, but also the sensitivity to probenecid of the outwardly rectifying currents triggered by the E protein, was an indication that these viral proteins trigger cellular alterations, such as morphological changes and cell death, that are inducers of pannexin-like current. Globally, these observations remain consistent with previous observations that both E and 3a proteins are mainly localized in intracellular compartments in various cell types [14,43,44,45,46,47]. Therefore, it is fair to mention that we cannot fully conclude the viroporin nature of these viral proteins, as their localization in subcellular organelles prevents us from clearly testing their intrinsic potential for channel activity.

We showed that expression of either of these two proteins in CHO cells induces an increase in cell death, as quantified by flow cytometry experiments. It is likely, although we did not investigate this point in detail, that this cell death accompanies the change in cell morphology and Petri dish detachment. As such, our observation that pannexin-like currents are mainly observed in detached round-shaped cells indicates that major cell morphology changes, up to the level of surface detachment, are required for the induction of pannexin-like currents. Whatever the exact mechanism, the upregulation of pannexin channels upon cell death has been previously observed [42]. It is thus not so surprising, in fact, that other reports faced problems reporting and identifying the conductances triggered by E and 3a viral proteins. The conditions for observing them are indeed quite drastic and require examining cells that are in the combined dying and detachment process, something that is not naturally pursued by researchers, especially if one hopes to detect viroporin activity. To reconcile our data with earlier publications, we noticed that whole-cell currents observed by others after SARS-CoV-1 E or 3a protein expression in HEK 293 cells [13,19] were also very similar to pannexin currents: outwardly rectifying current with a reversal potential close to 0 mV at physiological ion concentrations indicating poor ion selectivity and an amplitude of a few 100 pA.

One possibility is that the pannexin-like currents that we observed are due to the classical caspase-induced cleavage of pannexin [48]. Intriguingly, deletion of the C-terminal PBM of E protein abolished its pro-apoptotic effect, but cell morphology alteration and the induced outwardly rectifying currents were still present. Regarding the 3a protein, deletion of its PBM domain only decreased and did not completely abolish promoted cell death, but again, cell morphology alteration and pannexin-like currents were preserved. Altogether, these results suggest that cell morphology modification and pannexin induction may be linked and that these processes are not necessarily accompanied by cell death. One has to keep in mind that pannexin currents are activated by many stimuli in addition to cell death [48]. In particular, pannexin currents are also stretch-activated and may be enhanced in the detached cells that are undergoing major morphological alterations [26]. If pannexins are already activated by stretch, they would not be overactivated by their cleavage by caspase, which would explain the fact that E and 3a protein truncations do not prevent pannexin current, but only cell death. It may be difficult to clearly delineate the molecular nature of the outwardly rectifying currents in the absence of specific pharmacological tools [40,41,49]. Once the molecular nature of the CBX-sensitive currents is defined, it will be of interest to test if this channel is sensitive to the “viroporin” blockers that have been used elsewhere as evidence that the E/3a proteins are *bona fide* ion channels: amantadine, HMA, emodin, or xanthene [15,18,21].

## 5. Conclusions

In conclusion, SARS-CoV-2 native E and 3a proteins, and most likely SARS-CoV-1 ones as well, do not act as plasma membrane ion channels, but instead trigger the activity of plasma membrane pannexin channels, most likely through morphological alteration of the cells. However, our study does not rule out potential channel activity in intracellular membranes leading to morphological alterations and/or cell death. Additionally, adding a level of complexity, pannexin currents are associated with the induction of inflammation they may also be increased by cytokines such as TNF-alpha [50] and thus have been suggested as potential therapeutic targets [51,52,53]. Future studies will give more insights on the role of pannexin channels in COVID-19 physiopathology and treatment.

## Figures and Tables

**Figure 1 cells-12-01474-f001:**
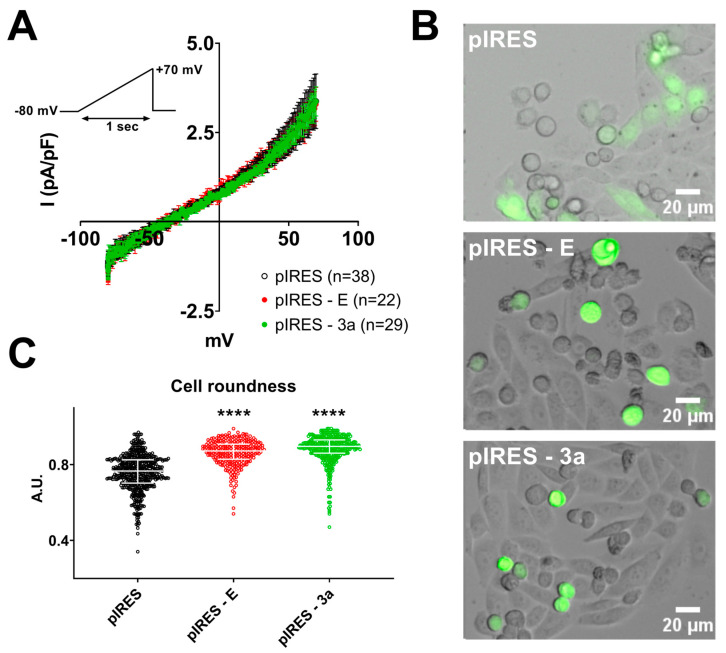
Expression of E or 3a protein is accompanied by altered cellular morphology but no modification in whole-cell currents in adhering cells. (**A**) Average current densities (±sem) recorded during the ramp protocol (inset) in adhering CHO cells expressing either only eGFP (pIRES), SARS-CoV-2 E and eGFP proteins (pIRES–E), or SARS-CoV-2 3a and eGFP proteins (pIRES–3a). (**B**) Superimposed brightfield and eGFP fluorescence images of CHO cells in the same 3 conditions as shown in A. (**C**) Morphology analysis, cf. Appendix A. Plot of individual cells (pIRES, n = 1922; pIRES–E, n = 1198; and pIRES–3a, n = 1283), median ± interquartile range. ****: *p* < 0.0001 as compared to pIRES control, Kruskal-Wallis test.

**Figure 2 cells-12-01474-f002:**
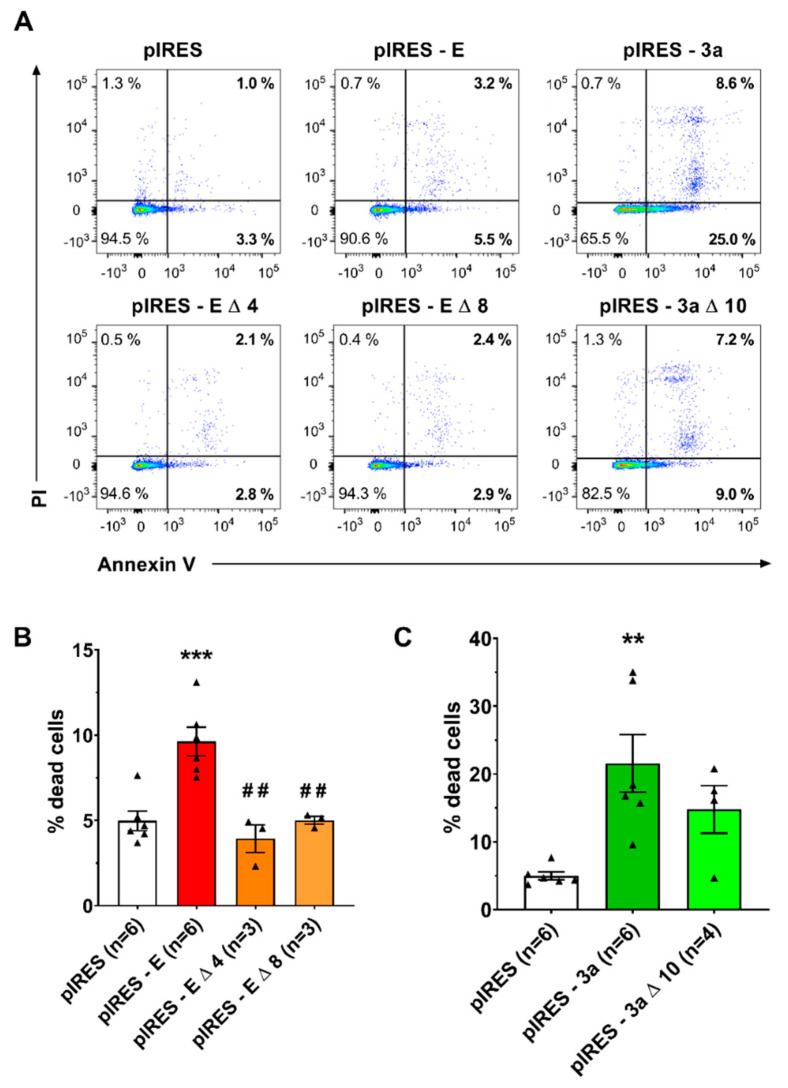
Expression of E and 3a protein induces cell death. (**A**) Flow cytometry analysis of eGFP-positive CHO cells expressing only eGFP (pIRES) or co-expressing eGFP and one of the following SARS-CoV-2 proteins: the full-length E protein (pIRES–E), C-terminally deleted E proteins (pIRES–E Δ 4 or pIRES–E Δ 8), full-length 3a protein (pIRES–3a) or C-terminally deleted 3a protein (pIRES–3a Δ 10). After 48 h of expression, cells were stained with annexin V AlexaFluor 647 (APC)/propidium iodide (PI, Perc-P). (**B**) Mean percentage (±sem) of Annexin V+ cells among eGFP-positive CHO cells expressing only eGFP (pIRES) or eGFP and full-length or truncated E protein. ***: *p* < 0.001 as compared to pIRES control, one-way ANOVA, ##: *p* < 0.01 as compared to E protein, *t*-test. (**C**) Mean percentage (±sem) of Annexin V+ cells among eGFP-positive CHO cells expressing only eGFP (pIRES), or eGFP and full-length or truncated 3a protein. **: *p* < 0.01 as compared to pIRES control, one-way ANOVA.

**Figure 3 cells-12-01474-f003:**
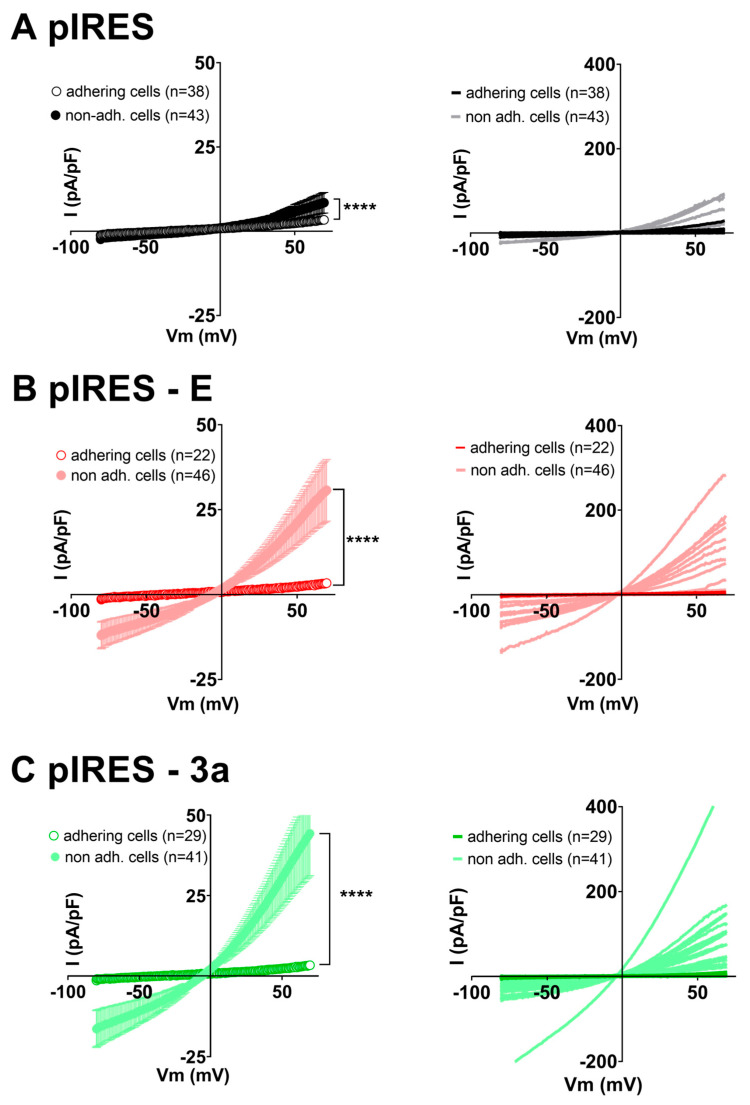
Expression of E or 3a protein is accompanied by outwardly rectifying currents in non-adhering CHO cells only. **Left,** average current densities (± sem) recorded during the ramp protocol in adhering (empty circles) or non-adhering (filled circles) CHO cells expressing either eGFP ((**A**), pIRES), SARS-CoV-2 E and eGFP proteins ((**B**), pIRES–E) or SARS-CoV-2 3a and eGFP proteins ((**C**), pIRES–3a). **Right,** plot of the individual adhering cells (darker color) or non-adhering cells (lighter color). ****: *p* < 0.0001, compared to adhering cells, two-way ANOVA on Ranks.

**Figure 4 cells-12-01474-f004:**
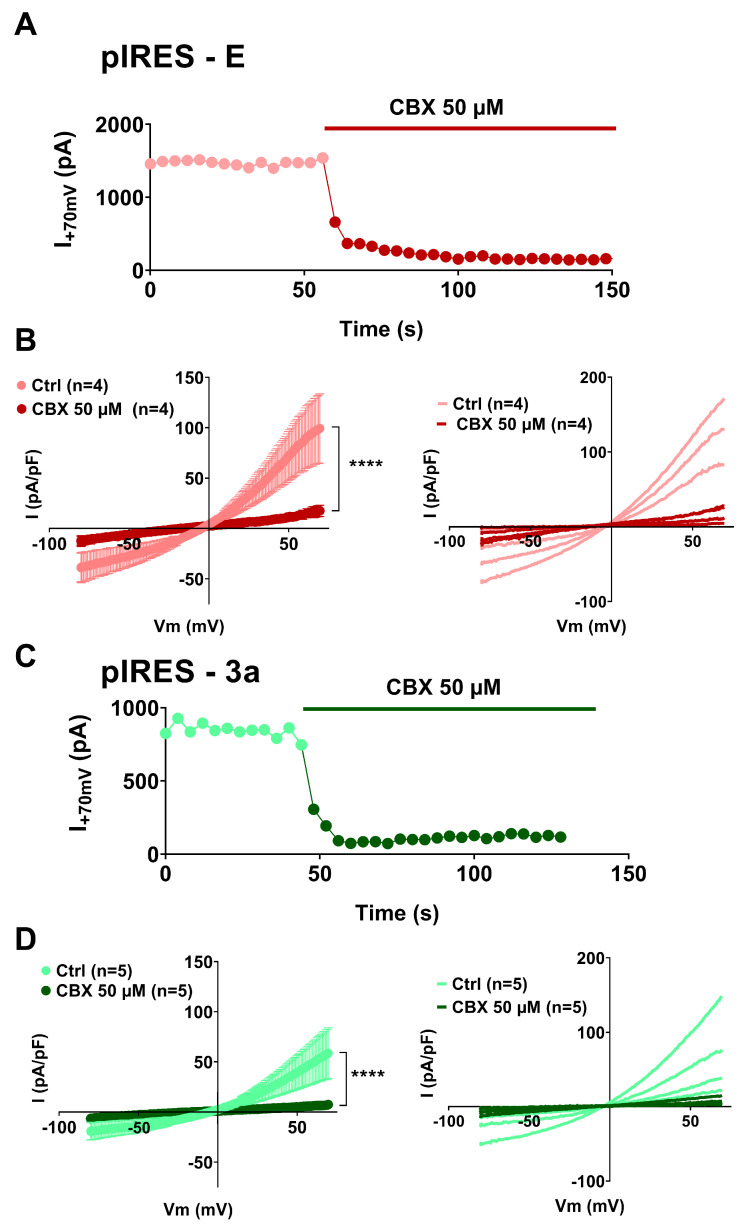
Currents due to E or 3a protein expression in non-adhering CHO cells are suppressed by the VRAC and pannexin inhibitor carbenoxolone (CBX). (**A**) Recording of the current amplitude in a CHO cell expressing SARS-CoV-2 E and eGFP proteins that displays large outwardly rectifying currents, in absence and presence of CBX. (**B**) **Left,** average current densities (±sem) recorded during the ramp protocol in non-adhering CHO cells, expressing SARS-CoV-2 E and eGFP proteins (pIRES–E), in absence (Ctrl, lighter color) and presence of CBX (darker color). **Right,** plot of the individual non-adhering cells in absence (lighter color) and presence of CBX (darker color). (**C**) Recording of the current amplitude in a CHO cell expressing SARS-CoV-2 3a and eGFP proteins that displays the outwardly rectifying currents, in absence and presence of CBX. (**D**) **Left,** average current densities (±sem) recorded during the ramp protocol in non-adhering CHO cells, expressing SARS-CoV-2 3a and eGFP proteins (pIRES–3a), in absence (Ctrl, lighter color) and presence of CBX (darker color). **Right,** plot of the individual non-adhering cells in absence (lighter color) and presence of CBX (darker color). ****: *p* < 0.0001, compared to Ctrl, two-way ANOVA on Ranks with repeated measures.

**Figure 5 cells-12-01474-f005:**
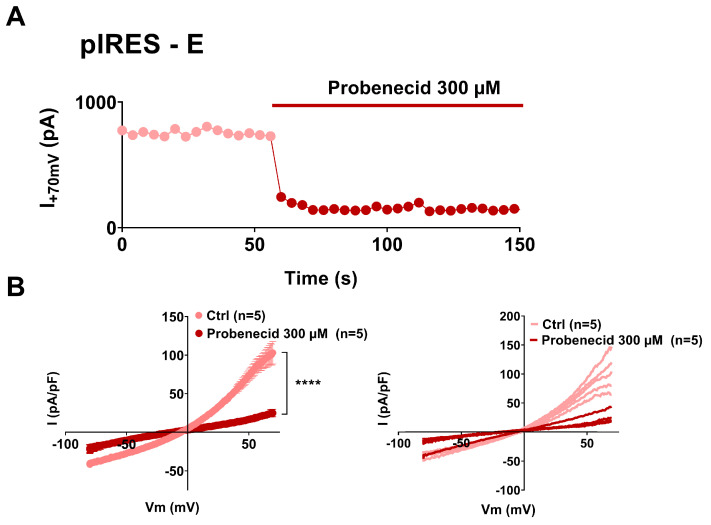
Currents due to E protein expression in non-adhering CHO cells are suppressed by the pannexin inhibitor probenecid. (**A**) Recording of the current amplitude in a CHO cell expressing SARS-CoV-2 E and eGFP proteins that displays large outwardly rectifying currents, in absence and presence of probenecid. (**B**) **Left**, average current densities (±sem) recorded during the ramp protocol in non-adhering CHO cells, expressing SARS-CoV-2 E and eGFP proteins (pIRES–E), in absence (Ctrl, lighter color) and presence of probenecid (darker color). **Right**, plot of the individual non-adhering cells in absence (lighter color) and presence of probenecid (darker color). ****: *p* < 0.0001 compared to Ctrl (DMSO), two-way ANOVA on Ranks with repeated measures.

**Figure 6 cells-12-01474-f006:**
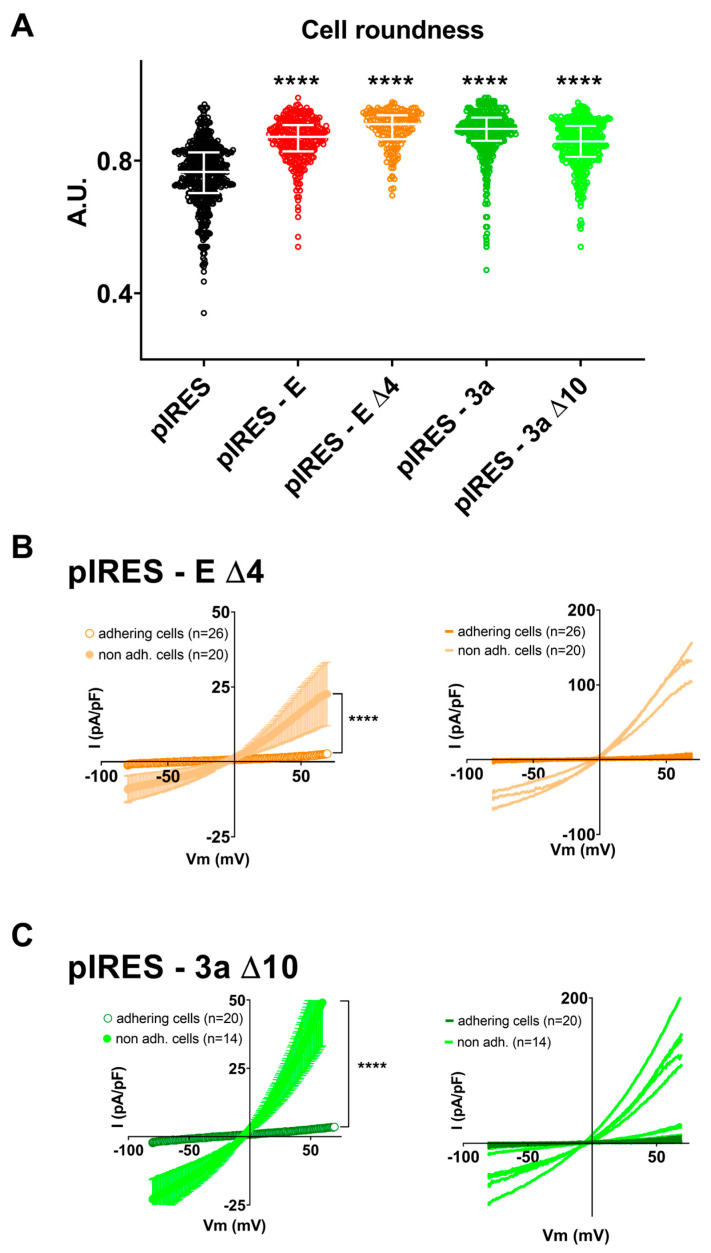
C-terminal deletion of E or 3a protein does not prevent cell alteration. (**A**) Cell roundness measured in CHO cells expressing eGFP (pIRES) alone or in combination with WT (pIRES–E) or the C-terminally deleted E protein (pIRES–E ∆4) or in combination with WT (pIRES–3a) or the C-terminally deleted 3a protein (pIRES–3a ∆10). Plot of individual cells, median ± interquartile range. ****: *p* < 0.0001 compared to pIRES control, Kruskal–Wallis test. (**B**) **Left,** average current densities (±sem) recorded during the ramp protocol in adhering (empty circles) or non-adhering (filed circles) CHO cells expressing the C-terminally deleted E protein (pIRES–E ∆4). **Right,** plot of the individual adhering (darker color) or non-adhering cells (lighter color). ****: *p* < 0.0001, compared to adhering cells, two-way ANOVA on Ranks. (**C**) **Left,** average current densities (± sem) recorded during the ramp protocol in adhering (empty circles) or non-adhering (filled circles) CHO cells expressing the C-terminally deleted 3a protein (pIRES–3a ∆10). **Right,** plot of the individual adhering (darker color) or non-adhering cells (lighter color). ****: *p* < 0.0001, compared to adhering cells, two-way ANOVA on Ranks.

## Data Availability

The data presented in this study are available on request from the corresponding author.

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
