# Peer review of "SARS-CoV-2 E and 3a Proteins Are Inducers of Pannexin Currents"

_cells, 2023, doi:10.3390/cells12111474_

Round 1
Reviewer 1 Report
In this study, Oliveira Mendes and colleagues investigate the consequences of heterologous expression of the SARS-CoV-2 E and 3a proteins in CHO cells. There has been much confusing and conflicting data recently reported on the potential viroporin activity of these proteins, and as pointed out by the authors, it is not clear whether these proteins indeed act as bona fide ion channels. Therefore, this study focuses on a highly significant topic which needs clarification. The studies reported are sound in methodology and are well described and presented. This study could be an important addition to the field. However, I feel it could be built upon to make a major contribution which would help to resolve outstanding controversy. I encourage the authors to take this opportunity to build on their current findings, and with additional experiments, complete a defining work which could be resubmitted here or elsewhere as a landmark study. Importantly, the current pharmacological evidence does not seem sufficient to conclusively point to pannexin channels as underlying the observed currents. That said, the current findings are sufficiently interesting to warrant publication – but the confidence in identification of pannexin channels as being responsible for the currents should be significantly moderated.
Here, the authors find that E and 3a expression increases cell roundedness and cell death – as determined by increased Annexin V+ staining and propidium iodide uptake. E/3a expression is associated with decreased cell adhesion and when non-adherent cells are subjected to whole-cell patch clamp analysis, increased outwardly-rectifying currents are observed, which are sensitive to carbenoxolone inhibition. Cells transfected with E-IRES-GFP or 3a-IRES-GFP plasmids which appear healthy do not display any significant currents beyond endogenous current observed in control cells.
This finding leads the authors to suggest that E and 3a induce changes in cell biology that result in secondary increases in plasma membrane ion channel activity and that E and 3a are not themselves plasma-membrane ion channels. Given the failure to consistently observe E/3a current amongst multiple groups, I suspect that this suggestion is correct. This is an important finding and is consistent with other careful electrophysiology experiments which also conclude that neither WT E or 3a act as plasma membrane channels (PMID: 33709461; PMID: 36695574).
The similarities in the properties of the conductance observed upon E and 3a overexpression adds weight to the conclusion that the conductance is not carried by either protein – but is a result of upregulation of the same native channel.
Findings without contention:
* Overexpression of both E and 3a in CHO cells increases cell death, and increased incidence of a carbenoxolone-sensitive conductance.
* The truncations do not affect the cell roundness observed upon E or 3a expression.
* Cell death is fully inhibited by delta4 truncation of E, and partially inhibited by delta10 truncation of 3a. BUT the increased conductance observed for full length E and 3a is not reduced by the same truncations. Therefore, the increased conductance is not correlated with cell death.
It appears likely that the increased conductance is not associated with cell death but in fact more closely associated with cell roundness. Based on this, I suggest that they de-emphasize cell death as the basis of the increased conductance, as currently included in the abstract and elsewhere (Lines 27-28: “Herein, we illustrate for the first time that carbenoxolone blocks these currents suggesting that they are conducted by pannexin channels, most likely activated by cell morphology change and/or cell death”), particularly as the increased conductances were not seen “in control dying cells” (line 97).
Proposed findings with contention:
The authors suggest that the increased conductance is carried by increased plasma-membrane expression of a pannexin channel. This study, while already useful, is an important opportunity to provide the field with a comprehensive characterization of the consequences of heterologous E/3a expression. If the authors conclusively show that E/3a over expression leads to increase plasma membrane expression of pannexin channels it would help to reconcile the controversies observed from many confusing published studies.
This conclusion would be greatly supported by more direct evidence of pannexin channel membrane expression. This could include immunofluorescence or Western Blot data.
It would be an important addition to the field to show whether the conductances observed here are sensitive to the “viroporin” blockers which have been used elsewhere as evidence that the E/3a proteins are bona fide ion channels. Are the outward rectifying currents seen here inhibited by amantadine, HMA, Emodin, Xanthene (PMID: 34853399)? I think it is possible that all the disparate data suggesting E/3a are plasma membrane ion channels is due to upregulation of native channels – which might have similar pharmacological profiles.
Given that it appears that the conductance is more closely related to cell roundness and not death, it seems possible that the increased conductance is a result of cell swelling. I suspect that the current might not be pannexin-mediated, but SWELL-1 (VRAC/LRCC8) currents.
The current profiles shown also resemble SWELL-1/VRAC/LRCC8 channel currents. In addition to actions on pannexin channels carbenoxolone also inhibits the other large-pore anion channel SWELL-1/VRAC/LRCC8 (PMID: 18837047, 19150332). Therefore, this single pharmacological effect cannot conclusively point to pannexin.
To distinguish pannexin currents from SWELL-1 currents, I suggest the authors conduct anion-substitution experiments, or profile a range of other pannexin inhibitors such as spironolactone (PMID: 29237722), glibenclamide, probenecid, and DIDS (PMID: 26755773). The peptide 10panx peptide (PMID: 19150332) could be used to more definitively identify pannexin channels as being responsible.
Given the potential dirty pharmacology of the large-pore anion channels, conclusive identification of the endogenous channel responsible via drug inhibition might be difficult, and plasma-membrane staining for pannexin or VRAC channel subunits would be highly informative.
Repeating studies in oft-used HEK293 cells would help to generalize the findings here to other contentious recent studies, but I appreciate this is beyond the immediate scope of the article.
Minor comments:
Ref 18 is a review by the same authors and should be changed to PMID: 34305855 where the primary data is reported.
How were cell counting experiments, described in lines 196-198, performed?
The choice of statistical tests are questionable: why use ANOVA in figs 3 and 5 – the p value here would, I think, just mean that any data point is significantly different to any other – it does not tell you which pairwise comparison is statistically significant. Journal editors should determine the validity of the statistical tests used.
Reviewer 2 Report
SARS-CoV-2 Envelope and Orf3a have been previously identified as viroporins that can form ion channels at the plasma membrane to facilitate the virus release from infected host cells. In this manuscript, authors used patch clamp data to demonstrate their findings that “SARS-CoV-2 E and 3a proteins are not viroporins at the plasma membrane.” Rather, the authors concluded that E and 3a proteins induce the activation of pannexin channels to mediate cell death.
Major comments:
Previous studies have demonstrated that 3a channels, like most ion channels, can express at both plasma membrane and cytoplasm. Strong evidence has supported both E and 3a proteins form ion channels not only in planar lipid bilayers and Xenopus oocytes, but also in mammalian HEK293 cells. For 3a proteins, confocal immuno-imaging found expression in both plasma membrane and cytoplasm [Chan CM, Tsoi H, Chan WM, Zhai S, Wong CO, Yao X, Chan WY, Tsui SK and Chan HY. The ion channel activity of the SARS-coronavirus 3a protein is linked to its pro-apoptotic function. Int J Biochem Cell Biol. 2009;41:2232-9.] This paper also revealed key amino acids that control the trafficking signal for efficient endoplasmic reticulum protein export for cell surface expression, deletion of these amino acids removed the 3a protein expression at the plasma membrane (Fig.1), which resulted in a significant reduction in current amplitude in IV curve comparable to the untransfected IV curve (Fig.6).
For E proteins, whole-cell patch clamp has recorded current from E proteins expressed in HEK293 cells [Pervushin K, Tan E, Parthasarathy K, Lin X, Jiang FL, Yu D, Vararattanavech A, Soong TW, Liu DX and Torres J. Structure and inhibition of the SARS coronavirus envelope protein ion channel. PLoS Pathog. 2009;5:e1000511.]. And the membrane currents were inhibited by hexamethylene amiloride (HMA). The stronger evidence for E protein expression at the plasma membrane forming ion channels is the finding that HMA, which was applied to bath solution, can bind to two binding sites in the E protein transmembrane domain.
Please include these in the discussion about the controversial studies about E/3a viroporins.
Additionally, the authors may be aware that cell detachment can alter the subcellular distribution of channel proteins.
Figures 3 and 4:
Q1: why holding Vm at +70mV? To obtain large current amplitude? Under physiological ion conditions, changes in currents around resting membrane potential are more physiologically relevant. At negative potentials, PANX1 channels have low conductance.
Q2: “Currents due to E and 3a protein expression in non-adhering CHO cells are suppressed by the pannexin inhibitor carbenoxolone (CBX).” Does this experiment suggest CHO cells have endogenous pannexin channels? Why CBX blockade not change the reversal potential on IV? In comparison, changes in 3a current amplitude in HEK293 are associated with changes in reversal potential [Chan CM, Tsoi H, Chan WM, Zhai S, Wong CO, Yao X, Chan WY, Tsui SK and Chan HY. The ion channel activity of the SARS-coronavirus 3a protein is linked to its pro-apoptotic function. Int J Biochem Cell Biol. 2009;41:2232-9.]. Please discuss the discrepancy.
Q3: what is CHO cell membrane capacitance? The currents in adhering cells expressing either E or 3a are small compared to that in non-adhering cells, but they are not small if Cm is typically around 20-50pF, representing E or 3a channel – generated currents. Cell detachment simply activated pannexin channels generating larger currents. How do you rule out this alternative possibility?
Minor comments
Line 405: “do no prevent”=>”do not prevent”
Round 2
Reviewer 2 Report
Authors have properly responded to my questions and comments.